# Non-Hermitian route to higher-order topology in an acoustic crystal

He Gao[1], Haoran Xue [2✉], Zhongming Gu [1], Tuo Liu [1], Jie Zhu [1,3✉] & Baile Zhang [2,4✉]

Topological phases of matter are classified based on their Hermitian Hamiltonians, whose real-valued dispersions together with orthogonal eigenstates form nontrivial topology. In the recently discovered higher-order topological insulators (TIs), the bulk topology can even exhibit hierarchical features, leading to topological corner states, as demonstrated in many photonic and acoustic artificial materials. Naturally, the intrinsic loss in these artificial materials has been omitted in the topology definition, due to its non-Hermitian nature; in practice, the presence of loss is generally considered harmful to the topological corner states. Here, we report the experimental realization of a higher-order TI in an acoustic crystal, whose nontrivial topology is induced by deliberately introduced losses. With local acoustic measurements, we identify a topological bulk bandgap that is populated with gapped edge states and in-gap corner states, as the hallmark signatures of hierarchical higher-order topology. Our work establishes the non-Hermitian route to higher-order topology, and paves the way to exploring various exotic non-Hermiticity-induced topological phases.

---

[1] Department of Mechanical Engineering, The Hong Kong Polytechnic University, Kowloon, Hong Kong SAR, China. [2] Division of Physics and Applied Physics, School of Physical and Mathematical Sciences, Nanyang Technological University, Singapore, Singapore. [3] The Hong Kong Polytechnic University Shenzhen Research Institute, Shenzhen, China. [4] Centre for Disruptive Photonic Technologies, Nanyang Technological University, Singapore, Singapore. ✉email: haoran001@e.ntu.edu.sg; jie.zhu@polyu.edu.hk; blzhang@ntu.edu.sg

Hermiticity lies at the foundation of quantum formulation, as it guarantees the real-valued eigenvalues and the orthogonality of eigenstates. These Hermitian properties help to define topology of quantum wave functions and allow for the classification of topological phases of matter[1–4]. For example, TIs can be classified into the ten Altland-Zirnbauer classes[5] based on their Hermitian Hamiltonian's symmetries. Topological invariants such as the Chern number[6] have been well established in Hermitian systems, determining the topological boundary states through the principle of bulk-boundary correspondence. In classical systems, many photonic and acoustic TIs have been proposed to emulate the properties of TIs, especially in the classical analogs of quantum Hall[7–10], quantum spin Hall[11–16] and quantum valley Hall[17–19] effects. While these classical topological systems follow the Hermitian topology definition, they are intrinsically non-Hermitian because of the presence of loss and/or gain. On one hand, non-Hermiticity challenges the fundamental topological classification[20–22] and bulk-boundary correspondence[23–32]. On the other hand, it has brought topological physics substantially closer to real applications, as evidenced in the recent TI lasers[33,34].

Higher-order TIs are a type of newly predicted topological phases with a hierarchy of nontrivial topology, which host topological boundary states at "boundaries of boundaries"[35–40]. As a typical example, the quadrupole higher-order TI[35,36] carries a nontrivial topology in its two-dimensional (2D) bulk; however, it does not support one-dimensional (1D) gapless edge states as in a conventional TI, but supports zero-dimensional (0D) corner states at corners. Different from the tradition that topology needs to be first understood in condensed matter systems, higher-order TIs are realized almost entirely in classical artificial structures[41–46]. This means non-Hermiticity has been an issue since the very beginning—while higher-order topology is defined under the Hermitian condition, almost all higher-order TIs are in non-Hermitian systems. In the current understanding, the role of intrinsic loss, which is non-Hermitian, is very limited and generally negative: it only makes the topological boundary states to decay, but cannot determine the band topology, since it enters the Hamiltonian as uniform on-site imaginary parts, which have no effect on the real part of the dispersion, nor the eigenvectors. Hence, it is natural to ask whether non-Hermiticity can play a more important role in higher-order TIs. Recent theories have answered this question positively[47–53], but there has not been any experimental demonstration up to date.

In this work, we present an experimental demonstration of a non-Hermitian route to higher-order topology in an acoustic crystal. In contrast to previous higher-order TIs based on Hermitian designs[41–46], here the higher-order topology is induced by deliberately introduced losses that are non-Hermitian. Depending on the configuration of losses, the induced bandgap can be either topological or trivial, which can be judged with the biorthogonal nested-Wilson-loop approach[52]. With a carefully designed loss configuration, we experimentally identify the loss-induced topological bandgap through local acoustic measurement, followed by the direct observation of gapped edge states and mid-gap corner states, all of which are typical features of the higher-order topology. As a comparison, a different loss configuration can induce a trivial bandgap, in which no in-gap mode has been observed. Our work thus provides the experimental demonstration of non-Hermiticity-induced higher-order topological phases.

## Results

### The implementation of acoustic quadrupole TI
We consider a quadrupole TI which is a typical higher-order TI. In the quadrupole TI, as illustrated in Fig. 1a, a quadrupole moment in the bulk first induces dipole moments on the edges, which in turn induce charges at the corners, forming hierarchical boundary states[35,36]. A minimal model for the quadrupole TI is proposed by Benalcazar, Bernevig, and Hughes (BBH), in which the quadrupole phase, being Hermitian, is achieved by coupling dimerization[35,36]. Recently, a modified non-Hermitian BBH model has been proposed, showing that gain and loss can also induce a quadrupole phase[52]. This is accomplished by adding an on-site imaginary potential configuration to the BBH model, as shown in Fig. 1b, where blue and red sites have imaginary on-site potentials of $\gamma_1$ and $\gamma_2$, respectively. Note that we have set all couplings to have the same strength, i.e., no coupling dimerization, such that the system is gapless in the Hermitian limit (see Supplementary Note 2 for tight-binding calculations). When $\gamma_{1,2}$ are unequal nonzeros, a bandgap opens and a quadrupole TI emerges. We note that the key ingredient to achieve this is the difference between the on-site imaginary parts of the red and blue sites, i.e., $\gamma_1 \neq \gamma_2$. Gain medium, whose implementation requires complex designs in acoustics[54,55], is not necessary here. With this insight, we construct an acoustic crystal to realize this tight-binding model with only losses. The designed unit cell (lattice constant $a = 400$ mm) is illustrated in Fig. 1c, which consists of 16 cuboid acoustic resonators of sizes 80 mm × 40 mm × 10 mm, coupled through identical thin waveguides of width 4 mm. The wall thickness of each resonator is 6 mm. The sign of a coupling can be chosen by the location of the thin waveguide[41,42,56,57] (see more details in Supplementary Note 3). Waveguides exhibiting positive and negative couplings are colored in yellow and gray in Fig. 1c, respectively. The resonators colored in blue only have a background loss that is intrinsic to the resonators; we set it as $\gamma_1$. Besides the intrinsic loss $\gamma_1$, the resonators colored in red have an additional loss $\gamma_2 - \gamma_1$, which is introduced by drilling small holes on the sidewalls of resonators and then filling these holes with acoustic absorbing materials (see Fig. 1d for a photo of the real structure where small holes sealed with black absorbing materials are clearly visible). This newly introduced loss turns the system from a gapless phase to a gapped phase (see Supplementary Note 2 for bulk bandstructure), which, as we will demonstrate both numerically and experimentally, is a topological quadrupole phase.

The loss-induced bulk quadrupole moment can be characterized in a similar way to the Hermitian case[35,36], but with a biorthogonal basis[52]. The hierarchy of the quadrupole topology can be revealed by two Wilson loops. The first Wilson loop calculation over all the bands below the bandgap gives Wannier bands that are symmetrically distributed with respect to atomic center, indicating the vanishing of bulk dipole moment. However, a second Wilson loop over a certain Wannier sector gives a quantized polarization of 1/2, which indicates that the edge Hamiltonian is a TI with a quantized dipole moment, which is induced by a quantized bulk quadrupole moment (see Supplementary Note 2 for more details). As a consequence of the nontrivial bulk quadrupole moment, gapped edge states and in-gap corner states should be found in a finite sample. To see this, we perform numerical calculations on a finite acoustic lattice and plot the resulted eigenfrequencies in Fig. 2a. Apart from the bulk states (gray dots), gaped edge states (yellow dots) and four degenerate in-gap corner states (red dots), which are induced by the bulk topology, are also found. We also plot the sum of probabilities (using right eigenvectors) of the bulk, edge, and corner states in Fig. 2b–d, respectively, further identifying their existence. We note, as can be seen from Fig. 2b–d, the states distribute all over the bulk, edges, and corners, and thus this system does not feature non-Hermitian skin effect.

### Experimental demonstrations in acoustic lattices
To demonstrate above phenomena experimentally, we fabricate an acoustic

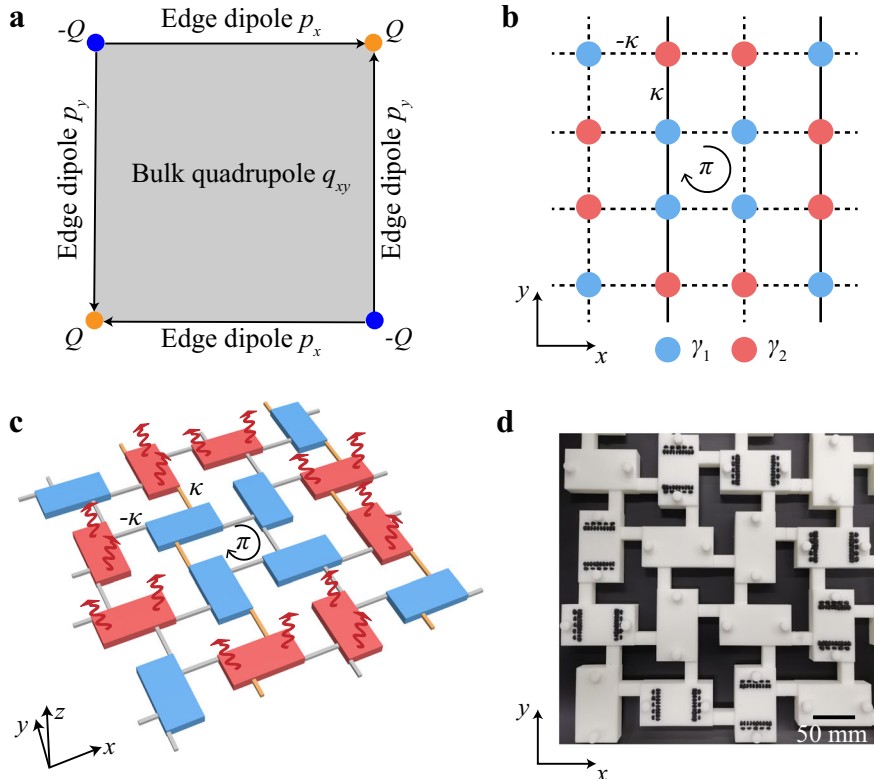

**Fig. 1 Non-Hermiticity-induced quadrupole topological insulator and its acoustic implementation. a** Schematic of a bulk quadrupole moment ($q_{xy}$) with its accompanying edge dipole moments ($p_x$ and $p_y$) and corner charges ($Q$). **b** Tight-binding model of a unit cell that consists of 16 sites. Red (blue) sites have an imaginary on-site potential of $\gamma_2$ ($\gamma_1$). Solid (dashed) lines denote positive (negative) couplings with strength $\kappa$. There is a $\pi$ flux threading each plaquette due to the negative couplings. **c** Acoustic design of the lattice model in (**b**). Blue and red cuboids represent acoustic resonators with only background loss ($\gamma_1$) and those with deliberately enhanced loss ($\gamma_2$), respectively. These resonators are coupled by identical thin waveguides. The coupling waveguides that exhibit positive (negative) couplings are colored in yellow (gray). **d** Photo of the fabricated sample that realizes the acoustic lattice shown in (**c**). Some resonators have small holes sealed with black absorbing materials to enhance the losses.

lattice through stereo-lithography 3D printing, with 12 resonators along each of $x$ and $y$ directions (see Fig. 3a for a photo of the sample). Each resonator has two small holes that can be opened or closed by two circular covers. In the experiment, the sound waves generated by a speaker are guided into the sample through a small hole at one side of a resonator. A microphone detects the signals through the other hole at the other side of the same resonator. This measurement is repeated for all the resonators of the sample (see "Methods" and Supplementary Note 4 for more details). We first focus on resonators in the bulk. According to Fig. 2a, there are two branches (one around 2116 Hz and the other around 2168 Hz) of bulk states separated by a real frequency gap. In each branch, the eigenstates are further divided into two sub-branches by an imaginary frequency gap. The states with imaginary parts around 20 Hz mainly distribute on the resonators without additional losses (blue sites in Fig. 1c), while those with imaginary parts around 80 Hz mainly distribute on the resonators with additional losses (red sites in Fig. 1c). Thus, the measured responses from bulk resonators with additional losses are very low in intensity, containing no useful information. However, the responses from bulk resonators without additional losses are relatively high, which can be used to characterize the bulk bandgap. Here we choose a bulk resonator at the center (labeled "3" in Fig. 3a) and plot its measured response spectrum as the blue curve in Fig. 3b. Two peaks can be clearly observed that correspond to the two branches of bulk states with longer lifetime (around imaginary 20 Hz). A similar situation applies to the resonators at edges without additional losses. We choose a

resonator in the middle of one edge (labeled "2" in Fig. 3a) and plot its response spectrum as the yellow curve in Fig. 3b. The two peaks correspond to the gapped edge states. In contrast to the bulk and edge spectra, the measured spectrum on a corner resonator (labeled "1" in Fig. 3a) only has one single peak located around 2142 Hz, as shown by the red curve in Fig. 3b, which is consistent with the predicted eigenfrequency of corner states (the spectra from other corners are similar and thus are not shown). To further demonstrate the non-Hermiticity-induced quadrupole phase, we also plot in Fig. 3c–e the site-resolved responses measured at peak frequencies of the corner, edge, and bulk spectra, respectively. At 2142 Hz which corresponds to the peak of the corner spectrum, the measured acoustic intensity at the corners is much higher than other regions (Fig. 3c), showing the existence of corner states. In contrast, the measured responses at the peak frequencies (2164 and 2170 Hz) for edge and bulk spectra are higher in the edge and bulk regions, respectively (Fig. 3d and e). We note that there is considerable overlap between the spatial maps for the edge states (Fig. 3d) and the bulk states (Fig. 3e) due to the fact that the edge states and bulk states are quite close in frequency. These experimental observations agree well with numerical simulations presented in Fig. 2 and Supplementary Note 6.

As a comparison, we further demonstrate that a different loss configuration can open a trivial bandgap. The unit cell for such a trivial lattice is shown in Fig. 4a. Although in this case additional losses still open a bandgap, the Wannier sector polarization is zero and only bulk states are found for a finite lattice (see Fig. 4b

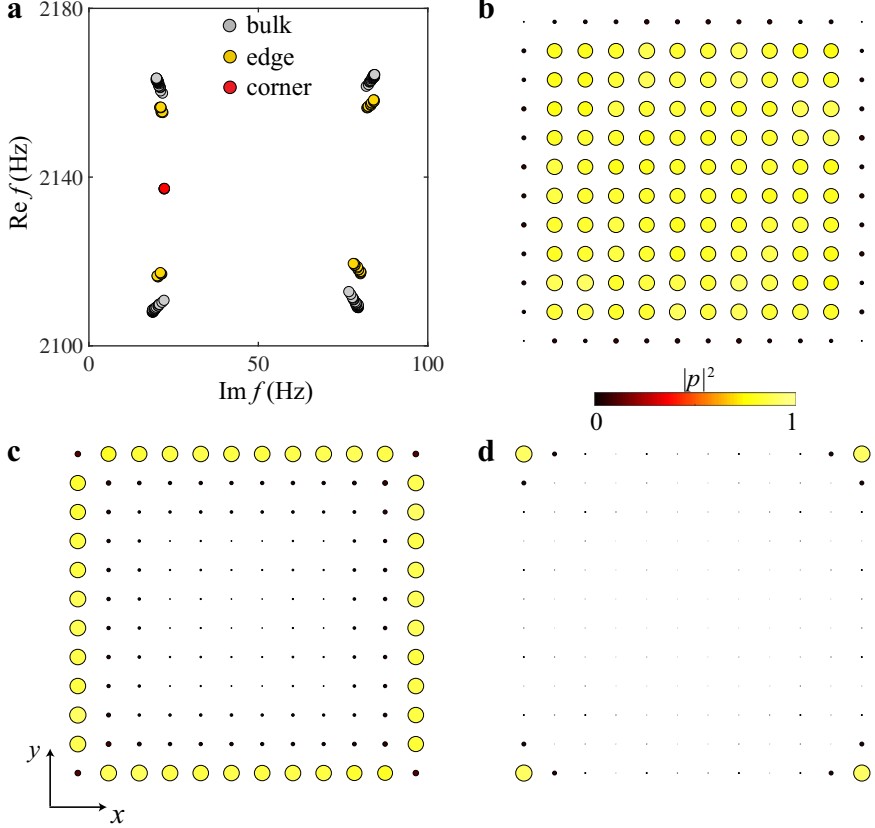

**Fig. 2 Eigenstates for a finite nontrivial lattice. a** Numerically calculated eigenfrequencies of the nontrivial lattice with 12 resonators along *x* and *y* directions. Gray, yellow, and red dots represent bulk, edge, and corner states, respectively. **b-d** Sum of probabilities for the bulk (**b**), edge (**c**), and corner states (**d**), respectively.

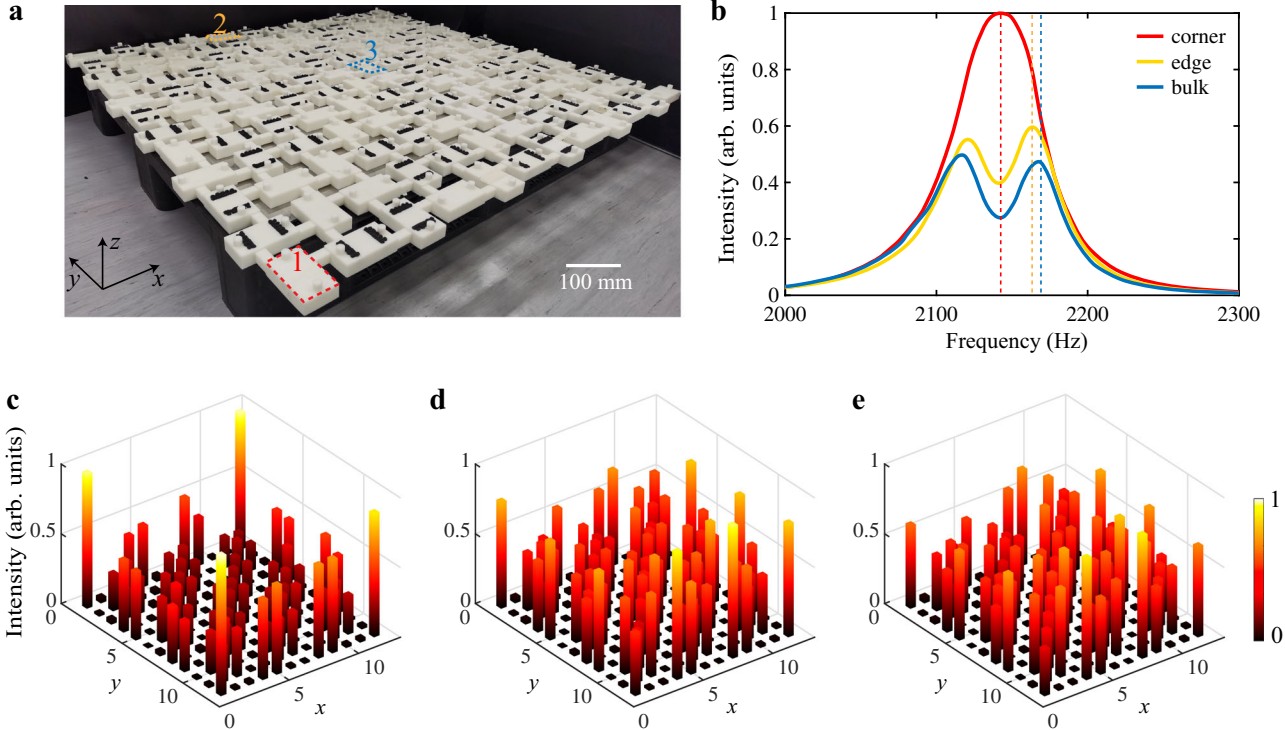

**Fig. 3 Experimental observation in the nontrivial lattice. a** Photo of a 3D printed lattice with 12 × 12 resonators. **b** Measured acoustic intensity spectra for the sample in (**a**). Red, yellow, and blue curves represent the corner, edge, bulk spectra measured at the resonators labeled as "1", "2", and "3" in (**a**), respectively. **c-e** Measured intensity profiles at the peaks of the corner, edge, bulk spectra, denoted by the red (2142 Hz), yellow (2164 Hz), and blue (2170 Hz) dashed lines in (**b**), respectively. In **c-e**, both the height and color of each bar indicate the power strength.

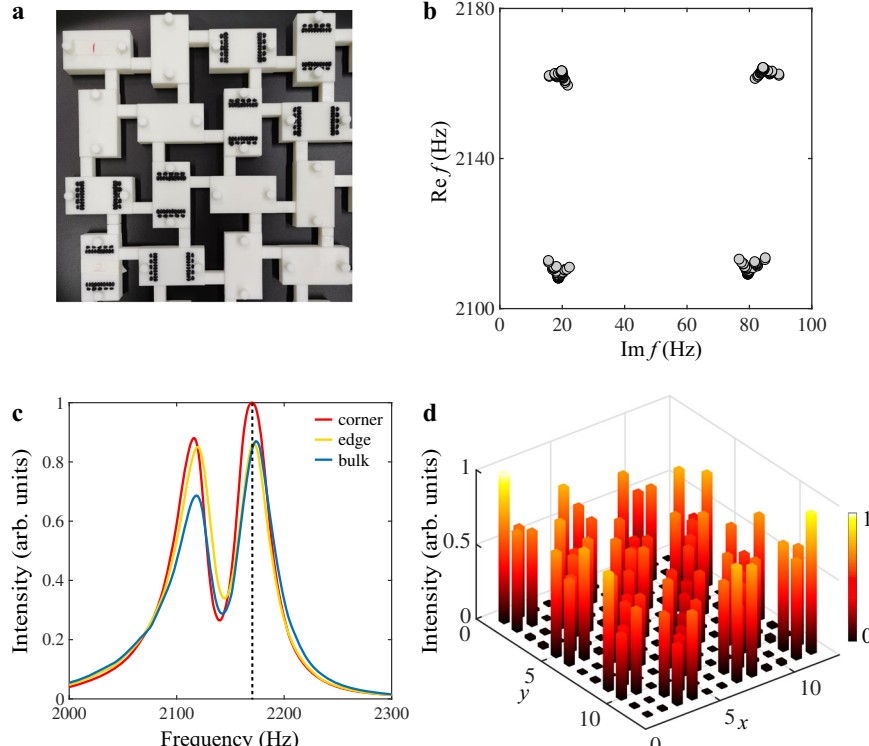

**Fig. 4 Experimental observation in the trivial lattice. a** Photo of one unit cell for the trivial lattice. **b** Simulated eigenfrequencies for the trivial sample with 12 resonators along $x$ and $y$ directions. **c** Measured spectra for the sample in (**a**). Red, yellow, and blue lines represent the spectra measured in one of resonators within the corner, edge, and bulk regions, respectively. **d** Measured intensity profile around the spectra peak denoted by the black dashed line in (**c**).

for eigenfrequencies for a finite lattice). We again conducted local acoustic measurements over all sites on a finite trivial lattice with the same sizes as the topological one. Measured response spectra for a resonator without additional losses in corner, edge, and bulk regions are plotted in Fig. 4c. In contrast to the topological lattice, now all three curves have two peaks which are located at frequencies corresponding to the bulk states. We further plot measured intensity around one of the peaks (2170 Hz) in Fig. 4d. As can be seen, the acoustic energy distributes over the whole lattice, indicating the peaks in Fig. 4c are results of the bulk states.

## Discussion

In conclusion, we have designed and experimentally demonstrated a non-Hermitian route to constructing higher-order TI in an acoustic crystal. Topological corner states are found upon introducing additional losses to an originally gapless system. A trivial insulator can also be created with a different loss configuration. These results show that, being contrary to the common negative perception, losses can play a not only positive but also a decisive role in forming topological states. Our work points to a direction beyond the conventional Hermitian framework of topological physics, and offers a unique platform to study various non-Hermiticity-induced topological phases. For example, while our work focuses on quadrupole phases, other types of higher-order TIs can be similarly induced on this platform. Moreover, recent studies[58–61] have shown that many phenomena are commonly found in both acoustics and photonics. The phenomena found in this work can also be extended to photonics where the control over loss and gain can be more flexible. With externally controllable loss and gain, it will be promising to construct actively reconfigurable devices using these non-Hermiticity-induced corner states.

## Methods

**Numerical simulations**. All the simulations were performed with COMSOL Multiphysics, pressure acoustics module. In all simulations, the density for background medium air is set to be 1.22 kg/m$^3$, and the real part of sound speed in air is set to be 342.34 m/s. The losses are taken into account by the imaginary part of sound speed $c$. By fitting the measured and simulated spectra of the single resonators via adjusting the sound speed, $c$ is set to be $342.34 + 2.23i$ m/s for the resonators only with background loss, and $c = 342.34 + 14.72i$ m/s for the resonators with additional losses. To calculate the band structures, periodic boundary conditions are used for outermost boundaries, while other boundaries are considered as hard boundaries. When calculating the eigenfrequencies of the finite lattices (Figs. 2a and 4b), all the boundaries are considered as hard boundaries.

**Experimental details**. Two small air holes ($r = 1.2$ mm) were drilled at two sides of each resonator, which allow for the signal input and output in experimental measurements. A lock-in amplifier (Zurich Instrument HF2LI) connected to a computer functioned as the signal generator and data acquisition system simultaneously. The incident acoustic waves were generated by a loudspeaker with a swept signal ranging from 2000 to 2300 Hz. The acoustic fields inside the resonators were measured by a 1/4-inch microphone (Brüel & Kjær, Type 4935) that was placed at one side of each resonator and then transferred to the lock-in amplifier via a conditioning amplifier (Brüel & Kjær, 64 NEXUS Type 2693A). For the site-resolved response measurements (Figs. 3d–f and 4d), the input source and microphone were placed at the two sides of the same resonator for each measurement, and the measurement was repeated for all resonators to obtain the site-resolved maps.

## Data availability

The data that support the findings of this study are available from the corresponding author upon reasonable request.

## Code availability

All numerical codes are available from the corresponding authors on reasonable request.

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

## Acknowledgements

This research is supported by the General Research Fund scheme of Research Grants Council of Hong Kong under Grant No. 15205219 and Singapore Ministry of Education Academic Research Fund Tier 3 under Grant No. MOE2016-T3-1-006 and Tier 2 under Grant No. MOE2018-T2-1-022(S).

## Author contributions

H.G., H.X., J.Z., and B.Z. conceived the idea. H.X., J.Z., and B.Z. supervised the project. H.G. and H.X. designed the samples and performed the simulations. H.G., Z.G., and T.L. conducted the experiments and analyzed the data. All authors contributed to the writing and discussions of the manuscript.

## Competing interests

The authors declare no competing interests.
