## [Peer Review File · Nature Communications]

Reviewers' Comments:

Reviewer #1:

Remarks to the Author:

This manuscript titled "Non-Hermitian route to higher-order topology in an acoustic crystal" presents a very interesting study about experimentally realizing higher-order topology, in a unique non-Hermitian way with acoustic system. The authors designed and fabricated a two-dimensional acoustic metamaterial lattice, demonstrating a quantized quadrupole moment induced by non-Hermiticity, much different from the Hermitian quadrupole TIs. The non-Hermiticity was achieved by introducing sound absorption at certain sites. It's impressive that the tight-binding model is experimentally implemented with a simple design, and 0D corner states, 1D edge states and 2D bulk states can be observed. The non-trivial and trivial topology can be switched by adjusting the positions of the lossy sites, which confirms their design.

The manuscript is well organized and written. The experimental results demonstrated the expected properties of a quantized quadrupole TI. The novelty meets the criteria of Nature Communications. At the same time, I do have some questions out of curiosity. Therefore, I would like to recommend the publication of this manuscript after the authors address the following questions:

- 1) In Hermitian quadrupole TI, generally it's 2×2 elements for a basic unit cell. Why did the authors choose 4×4 elements arrangement?
- 2) Could the authors comment on some possible ways to widen the non-Hermiticity induced bandgap?
- 3) The coupling waveguides are kind of narrow, the authors need to discuss the lossy effect in the coupling channels.
- 4) The acoustic energy distributions for the resonators in dipole mode are inhomogeneous, so how did the authors define the acoustic intensities in Fig. 3 and 4? The results are the average values in the whole resonator or the values at a certain point?

Reviewer #2:

Remarks to the Author:

In this manuscript, the authors reported the experimental realization of a non-Hermitian higher-order topological insulator in an acoustic crystal. They considered a quadrupole topological insulator model with the Hermitian part in a gapless phase, and the nontrivial topology was induced by deliberately introduced losses. Using local acoustic measurements, the authors identified the topological bulk and edge band gaps as well as the in-gap corner states. This work establishes the non-Hermitian route to higher-order topology.

Overall, I think this work is timely and important. It would be useful if the authors could give some discussions on the following points:

1. Is it principally possible to introduce gain in such acoustic systems?
2. The background loss γ_1 is comparable with the coupling strength κ in their setup. In experiment, can the ratio (γ_1 over κ) be made very small and what are the limitations? A smaller ratio would narrow the relative linewidth of the transmission measurement, which may allow the extraction of more band/state information.
3. Related to the above point, the measured responses from γ_2 resonators contain no useful information due to significant loss. However, in the weak γ_1 limit, can the band gap be observed from the responses of γ_2 resonators? The authors should provide some simulations if possible.
4. Related to the non-Hermiticity, it is possible to generate asymmetric tunneling (as in Ref. 41) for such acoustic systems?

Reviewer #3:

Remarks to the Author:

Referee Report for: "Non-Hermitian route to higher-order topology in an acoustic crystal".

manuscript by Dr Zhang and co-workers

=====

In this interesting work, the authors report an experimental demonstration of the loss-induced higher-order topological phase (i.e., non-Hermitian higher-order topological insulator) in an acoustic crystal. Hermitian higher-order topological insulators, as a new type of topological phases with a hierarchy of nontrivial topology, have been considerably investigated in both theories and experiments, while non-Hermitian higher-order topological phases have only been theoretically proposed.

As far as I know, this is the first experimental study of the non-Hermitian higher-order topological phase. The authors experimentally identify the loss-induced topological bandgap, depending on the loss configurations, and report the gapped edge states and mid-gap corner states. The manuscript is clearly presented and it should be appealing to a broad readership.

As a first experimental work of non-Hermiticity-induced higher-order topological phases, I think this work is timely and important. Moreover, there is a growing (but still relatively small) number of papers exploiting analogies between optical and acoustic phenomena, with these type of approaches and ideas. Therefore, I would like to recommend this work for publication after the following issues are well addressed in the actual manuscript (not just in the reply, which is not accessible to readers).

1. By comparing Fig.3(d) with 3(e), I cannot see a big difference between them. Especially, the spectrum in Fig.3(d) shows a strong acoustic intensity in the bulk regime, preventing us from inferring it is an edge excitation spectrum. Thus, the author should clarify this, and show a strong evidence it is an edge excitation.
2. Can the authors simulate the spatial intensity distribution of corner, edge and bulk excitations using COMSOL? Then readers can make a clearer comparison with the experimental results in Figs. 3(c-e), thus have a better understanding.
3. The authors should discuss how the distribution of disorder/defects and their strength may affect the robustness of loss-induced corner states. T
4. The authors consider two kinds of loss configurations, where one induces a topological quadrupole insulator, and the other leads to a trivial insulator. Can the authors provide us a clear physical picture why one of loss configurations can lead to the higher-order topological phase, and the other cannot?
5. The authors might want to refer to these closely related works: Nature Communications 10, 580 (2019); Phys. Rev. Lett. 123, 054301 (2019); Phys. Rev. Lett. 118, 040401 (2017).
6. The authors are using analogies between optics and acoustics, and these have been recently (2019 and 2020) explored in depth in the following works: Transverse spin and surface waves in acoustic metamaterials, Phys. Rev. B 99, 020301(R) (2019); Spin and orbital angular momenta of acoustic beams, Phys. Rev. B 99, 174310 (2019); Acoustic versus electromagnetic field theory: scalar, vector, spinor representations and the emergence of acoustic spin, New Journal of Physics 22, 053050 (2020); Edge modes in two-dimensional electromagnetic slab waveguides: Analogs of acoustic plasmons, Phys. Rev. B 102, 045129 (2020). I am not requesting all of these to be cited, but it seems

that the authors better to be aware of them, because what they are doing is importing ideas from optics and CM into acoustics, and this is what these recent works also do.

Response Letter to Reviewers

We are grateful for the constructive comments on this manuscript (NCOMMS-20-26559-T) from all the reviewers. In the text below the comments from each reviewer are quoted in *blue italics*, and followed by our response. We have also revised the manuscript and the Supplementary Information accordingly, and these updates are highlighted in **red** in those files. In the text below, references to these updates are also highlighted in **red**.

Reviewer #1

Reviewer comments:

This manuscript titled “Non-Hermitian route to higher-order topology in an acoustic crystal” presents a very interesting study about experimentally realizing higher-order topology, in a unique non-Hermitian way with acoustic system. The authors designed and fabricated a two-dimensional acoustic metamaterial lattice, demonstrating a quantized quadrupole moment induced by non-Hermiticity, much different from the Hermitian quadrupole TIs. The non-Hermiticity was achieved by introducing sound absorption at certain sites. It’s impressive that the tight-binding model is experimentally implemented with a simple design, and 0D corner states, 1D edge states and 2D bulk states can be observed. The non-trivial and trivial topology can be switched by adjusting the positions of the lossy sites, which confirms their design. The manuscript is well organized and written. The experimental results demonstrated the expected properties of a quantized quadrupole TI. The novelty meets the criteria of Nature Communications. At the same time, I do have some questions out of curiosity. Therefore, I would like to recommend the publication of this manuscript after the authors address the following questions

Response:

We thank Reviewer #1 for the encouraging comments. Below we provide point-to-point response to reviewer’s comments. We have also revised the manuscript following the reviewer’s suggestions.

Reviewer comments:

1) In Hermitian quadrupole TI, generally it’s 2×2 elements for a basic unit cell. Why did the authors choose 4×4 elements arrangement?

Response:

We thank Reviewer #1 for this insightful question. The reason why we choose a unit cell with 16 resonators can be understood intuitively with an effective dimerization picture. In the Hermitian case, the coupling dimerization opens the bulk bandgap. In our non-Hermitian approach, the losses induce an effective dimerization. To see this, let's first consider a simple toy model which consists of two coupled resonators with coupling strength κ . One of the resonators has loss of γ (see the inset of Fig. R1a). Fig. R1a shows real parts of the eigenfrequencies against the loss parameter γ . As can be seen, the difference between the real parts of the two eigenfrequencies decreases as γ increases. This can be understood as an effective reduction of the coupling strength between the two resonators. In contrast, if the two resonators have the same dissipation, the real parts of the eigenfrequencies will not be affected. According to this interpretation, the non-Hermitian unit cell with 16 resonators (Fig. R1b) effectively corresponds to a lattice with coupling dimerization as shown in Fig. R1c, which satisfy the dimerization pattern in the topological quadrupole lattice (BBH model) given in [*Science* **357**, 61-66 (2017)]. However, if we choose a smaller unit cell for the non-Hermitian lattice, we cannot realize the proper effective coupling dimerization. We would like to note that this effective dimerization picture has also been used to explain the 1D non-Hermitian topological chain with four (instead of two) resonators in one unit cell (see [*Phys. Rev. Lett.* **121**, 213902 (2018)]).

Fig. R1. Effective dimerization induced by loss. **a** Real part of eigenvalues for a simple dimer model. The inset shows the details of the dimer: two coupled resonators with coupling strength κ and one of the resonators has loss of γ . **b** Unit cell of the non-Hermitian lattice in our work, which can be effectively mapped to a lattice with coupling dimerization shown in **c**.

Reviewer comments:

2) Could the authors comment on some possible ways to widen the non-Hermiticity induced bandgap?

Response:

The bandgap can be widened by increasing the coupling strength κ and the additional loss γ_2 . For example, in our current design, $\kappa=17.08$ Hz and $\gamma_2 = 0.0415$, and the induced bandgap size is 45 Hz. If we increase κ and γ_2 to be 42.7 Hz and 0.083 respectively, then the bandgap becomes 120 Hz.

Another approach to making the bandgap wider is to go beyond the tight-binding model. In the tight-binding design, we have to make the coupling strength relatively small so that the real structure can be accurately mapped to desired tight-binding model. If we can realize the non-Hermiticity-induced topological bandgap in a phononic crystal design which goes beyond coupled-mode description, the bandgap would be much larger. This is also one important direction for future work.

We have incorporated above discussions into Supplementary Note 2 as “A larger bandgap can be achieved by increasing coupling strength κ and additional loss γ_2 . Moreover, extending the design to phononic crystal structures that go beyond tight-binding model may also yield a larger bandgap.”

Reviewer comments:

3) The coupling waveguides are kind of narrow, the authors need to discuss the lossy effect in the coupling channels.

Response:

We thank Reviewer #1 for this insightful suggestion. As pointed out by the reviewer, the narrow connecting waveguides would unavoidably cause acoustic wave energy dissipation. To check the lossy effect in the coupling channels, we performed full-wave simulations on structures with different amount of loss in the coupling channels. As can be seen in Fig. R2, the bandstructures almost remain the same as we increase the loss in the coupling channels from γ_1 (a) to $2\gamma_1$ (b) and $3\gamma_1$ (c). Here γ_1 is the background loss in the resonators. Thus, the extra loss in the coupling channels will not affect the bulk topology. We have added a new section “Supplementary Note 5” in the revised Supplementary Information to address this issue.

Fig. R2. Bandstructures for lattices with different losses in coupling waveguides. a-c, Bulk dispersion for acoustic lattices with loss in the coupling waveguides being γ_1 (a) to $2\gamma_1$ (b) and $3\gamma_1$ (c).

Reviewer comments:

4) *The acoustic energy distributions for the resonators in dipole mode are inhomogeneous, so how did the authors define the acoustic intensities in Fig. 3 and 4? The results are the average values in the whole resonator or the values at a certain point?*

Response:

We thank Reviewer #1 for pointing out this issue. The sound energy is measured at the side of each resonator where the intensity is maximal. As explained in the Methods part, two small holes are drilled at two sides of each resonator for signal excitation and detection. The microphone is inserted into one of the holes to collect the signal during the measurements. To clarify this issue, we have revised the Methods part as “**The acoustic fields inside the resonators were measured by a 1/4-inch microphone (Brüel & Kjør, Type 4935) that was placed at one side of each resonator and then transferred to the lock-in amplifier via a conditioning amplifier (Brüel & Kjør, 64 NEXUS Type 2693A).**”

Reviewer #2

Reviewer comments:

In this manuscript, the authors reported the experimental realization of a non-Hermitian higher-order topological insulator in an acoustic crystal. They considered a quadrupole topological insulator model with the Hermitian part in a gapless phase, and the nontrivial topology

was induced by deliberately introduced losses. Using local acoustic measurements, the authors identified the topological bulk and edge band gaps as well as the in-gap corner states. This work establishes the non-Hermitian route to higher-order topology. Overall, I think this work is timely and important. It would be useful if the authors could give some discussions on the following points

Response:

We thank Reviewer #2 for the positive remarks. Below we provide point-to-point response to reviewer's comments. We have also revised the manuscript accordingly.

Reviewer comments:

1. Is it principally possible to introduce gain in such acoustic systems?

Response:

Yes, in principle, gain can be realized in this coupled acoustic resonators system. Till now, several works have realized artificial acoustic gain in one-dimension systems with the aid of external power, which include loudspeaker connected to a controller ([*Nat. Phys.* **14**, 942-947 (2018)]) and digitally virtualized atoms ([*Nat. Commun.* **11**, 251 (2020)]). These schemes can principally be extended to two-dimensional systems, but the design and implementation would be more complex. If gain can be practically applied, it will greatly enhance the performance of these non-Hermitian topological systems and offer more possibilities to construct actively reconfigurable devices. Thus, we believe this is a promising direction for future studies. To address this question, we have added one sentence on the right column of page 2 in the main text as “Gain medium, whose implementation requires complex designs in acoustics,”.

Reviewer comments:

2. The background loss γ_{1} is comparable with the coupling strength κ in their setup. In experiment, can the ratio (γ_{1} over κ) be made very small and what are the limitations? A smaller ratio would narrow the relative linewidth of the transmission measurement, which may allow the extraction of more band/state information.

Response:

We thank Reviewer #2 for the insightful comments. Since the background loss γ_{1} mainly comes from intrinsic material absorption, it is very hard to be further reduced in this design.

In literature, the background losses of similar coupled acoustic resonator systems are all similar. Here we list a few examples which show measured resonance peaks with similar quality factor: Fig. S3 in [*Nat. Mater.* **18**, 113-120 (2019)], Fig. 3b in [*Phys. Rev. Lett.* **122**, 244301 (2019)] and Fig. 2d in [*Phys. Rev. Lett.* **121**, 085702 (2018)].

On the other hand, the coupling strength κ can be tuned by changing the size of the coupling waveguides. However, the strong coupling will cause undesired effects like coupling between the mode of our interest with other modes and large resonance frequency shift, which drive the lattice away from the ideal tight-binding model. This issue is commonly encountered when people want to design coupled resonator lattice to realize a tight-binding model (see [*Nat. Mater.* **17**, 323-328 (2018)]).

Therefore, the ratio γ_1/κ in the current design is challenging to be made very small in experiment. We do agree that a small ratio is good for experimental measurements. Relating this to the first question raised by the reviewer, we note that introducing gain into the system can also improve the linewidth and allow for extraction of more information from experiment.

Reviewer comments:

Related to the above point, the measured responses from gamma_2 resonators contain no useful information due to significant loss. However, in the weak gamma_1 limit, can the band gap be observed from the responses of gamma_2 resonators? The authors should provide some simulations if possible.

Response:

Although the background loss γ_1 is comparable to the coupling strength, it is still much smaller than the additional loss γ_2 . Thus, the measured responses from γ_2 resonators are mainly determined by γ_2 rather than γ_1 . To see this, we perform simulations to obtain the spectra for γ_2 resonators for different values of γ_1 . The results are given by the dashed curves in Fig. R3. As can be seen, the acoustic pressure in γ_2 resonators is in general very low, regardless of the values of γ_1 . In real measurements, the measured curves in γ_2 resonators will also suffer from experiment noises and thus become useless. In contrast, the spectra for γ_1 resonators benefit from the reduction of γ_1 . As can be seen from the solid curves in Fig. R3, as γ_1 decreases, the linewidth of spectrum peak becomes narrower.

Fig. R3. Simulated spectra under different background loss. Simulated responses for resonators (in the bulk) with additional loss (dashed curves) and without additional loss (solid curves). Different colors represent different values of background loss.

Reviewer comments:

4. Related to the non-Hermiticity, it is possible to generate asymmetric tunneling (as in Ref. 41) for such acoustic systems?

Response:

To best of our knowledge, so far there are no proposals for realizing asymmetric couplings for such acoustic systems. However, there are proposals for photonic systems which can also be used to generate asymmetric couplings in acoustics. Here we show one possible design based on the idea from [*Phys. Rev. B* **92**, 094204 (2015)] and [*Phys. Rev. Research* **2**, 013280 (2020)]. Consider two acoustic ring resonators coupled by a linking ring (Fig. R4). For the clockwise mode, the hopping from the left resonator to the right one goes through the lower part of the linking ring. In contrast, the hopping from right to left goes through the upper part of the linking ring. Thus, we can introduce gain and/or loss to the linking ring to generate asymmetric coupling, as shown in Fig. R4. In practice, the loss can be introduced by absorbing materials like in our experiment while the gain can be implemented through external power as described above.

Fig. R4. Schematic for realizing asymmetric coupling in acoustics using ring resonators. Two acoustic ring resonators (site ring) are coupled through another ring (linking ring). Asymmetric coupling can be generated by putting gain and loss into the link ring.

Reviewer #3

Reviewer comments:

In this interesting work, the authors report an experimental demonstration of the loss-induced higher-order topological phase (i.e., non-Hermitian higher-order topological insulator) in an acoustic crystal. Hermitian higher-order topological insulators, as a new type of topological phases with a hierarchy of nontrivial topology, have been considerably investigated in both theories and experiments, while non-Hermitian higher-order topological phases have only been theoretically proposed.

As far as I know, this is the first experimental study of the non-Hermitian higher-order topological phase. The authors experimentally identify the loss-induced topological bandgap, depending on the loss configurations, and report the gapped edge states and mid-gap corner states. The manuscript is clearly presented and it should be appealing to a broad readership.

As a first experimental work of non-Hermiticity-induced higher-order topological phases, I think this work is timely and important. Moreover, there is a growing (but still relatively small) number of papers exploiting analogies between optical and acoustic phenomena, with these type of approaches and ideas. Therefore, I would like to recommend this work for publication after the following issues are well addressed in the actual manuscript (not just in the reply, which is not accessible to readers).

Response:

We thank Reviewer #3 for the high opinions. Below we provide point-to-point response to reviewer's comments. We have also revised the manuscript accordingly.

Reviewer comments:

1. *By comparing Fig.3(d) with 3(e), I cannot see a big difference between them. Especially, the spectrum in Fig.3(d) shows a strong acoustic intensity in the bulk regime, preventing us from inferring it is an edge excitation spectrum. Thus, the author should clarify this, and show a strong evidence it is an edge excitation.*

Response:

We agree with Reviewer #3 that the measured edge spectrum and bulk spectrum do not show a strong difference. This is because the edge states and bulk states are quite close in frequency. As can be seen from Fig. 2a in the main text, the separation between the edge states and bulk states is only around 8 Hz. This leads to a large overlap between the measured edge spectrum and bulk spectrum. However, we note that although there is considerable intensity in the bulk for Fig. 3d, the intensity on the edges is in general higher. For Fig. 3e, the bulk intensity is similar to the edge intensity. Thus, there is still a weak contrast between Fig. 3d and Fig. 3e. Similar comparison can also be seen from simulation results given in Fig. R5c and Fig. R5d. To address this issue, we have added one sentence on the left column of page 4 in the main text: “We note that there is considerable overlap between the spatial maps for the edge states (Fig. 3d) and the bulk states (Fig. 3e) due to the fact that the edge states and bulk states are quite close in frequency.”

Reviewer comments:

2. *Can the authors simulate the spatial intensity distribution of corner, edge and bulk excitations using COMSOL? Then readers can make a clearer comparison with the experimental results in Figs. 3(c-e), thus have a better understanding.*

Response:

We thank Reviewer #3 for the constructive suggestion. We have performed COMSOL simulations to obtain the spectra for corner, edge and bulk excitations (Fig. R5a) and also the spatial intensity distributions (Fig. R5b-d). These results are consistent with the experimental results given in Fig. 3 in the main text. We have added these simulation results into Supplementary Note 6.

Reviewer comments:

3. *The authors should discuss how the distribution of disorder/defects and their strength*

Fig. R5. Simulations on the nontrivial lattice. **a**, Simulated acoustic intensity spectra for the nontrivial sample. Red, yellow and blue curves represent the corner, edge, bulk spectra calculated at one of the resonators in the corner, edge and bulk. **b-d**, Simulated intensity profiles at the peaks of the corner, edge, bulk spectra, denoted by the red (2137 Hz), yellow (2157 Hz) and blue (2165 Hz) dashed lines in **a**, respectively.

may affect the robustness of loss-induced corner states.

Response:

We thank Reviewer #3 for the suggestion. We have performed numerical calculations based on tight-binding model as well as real structures to study the robustness of the corner states. We consider on-site resonant frequency perturbations which should be the main source of disorder in the experiment due to fabrication error and temperature fluctuation. Here two cases are studied: disorder on all sites except the four corners (case 1) and disorder only on the four corners (case 2). In the tight-binding calculations, the disorder on site i is introduced by changing the on-site terms as $f_0(1 + df_i)$, where f_0 is the resonant frequency without disorder and df_i are random numbers uniformly distributed from $-\delta$ to δ with δ being the disorder strength. For case 1, we found that the corner states remain stable as

long as the bandgap remains open (Fig. R6a). For case 2, the frequencies of the corner states shift and may coincide with bulk bands for large disorder strength (Fig. R6b). In both cases, the corner states survive for weak disorders. These behaviors are similar to what are found for the Hermitian case ([*Nat. Mater.* **18**, 108-112 (2019)], [*Nat. Mater.* **18**, 113-120 (2019)] and [*Nat. Photon.* **13**, 692-696 (2019)]). We also confirm above results through full-wave simulations. In the simulation, the disorder is introduced by placing a hard cylinder with random size inside the resonator (see the inset of Fig. R6c). The simulated eigenfrequencies for case 1 and case 2 are shown in Fig. R6c and Fig. R6d, respectively. As can be seen, the corner states persist to exist and the eigenfrequencies are no longer degenerate for case 2, which are consistent with tight-binding calculations. **We have incorporated above discussions into Supplementary Note 7 in the revised version.**

Fig. R6. Robustness of the corner states. **a-b**, Numerically calculated eigenfrequencies using tight-binding model for nontrivial lattices with different onsite disorder strengths. In **a**, the disorder is introduced to all sites except the four corners. In **b**, the disorder is only introduced to the four corners. **c-d**, Simulated eigenfrequencies for finite nontrivial acoustic lattices with onsite disorder on all sites except the four corners (**c**) and only on the four corners (**d**).

Reviewer comments:

4. *The authors consider two kinds of loss configurations, where one induces a topological quadrupole insulator, and the other leads to a trivial insulator. Can the authors provide us a clear physical picture why one of loss configurations can lead to the higher-order topological phase, and the other cannot?*

Response:

This question is closely related to the question raised by Reviewer #1 on why we need a large unit cell with 16 sites, and can also be understood with the effective dimerization picture. In the Hermitian case (BBH model in [*Science* **357**, 61-66 (2017)]), whether the lattice is topological or trivial is determined by the coupling dimerization configuration. When the intra-cell coupling is larger (smaller) than the inter-cell ones, the system is in the trivial (topological) phase. In the non-Hermitian case, the coupling between two sites with different losses is effectively reduced and thus the loss configuration leads to effective coupling dimerization as described in Fig. R1. For the loss configuration that leads to the topological quadrupole phase (Fig. R7a), the corresponding effective coupling dimerization configuration is shown in Fig. R7b, which corresponds to the topological phase of BBH model. In contrast, the other loss configuration (Fig. R7c) corresponds to the trivial coupling dimerization configuration (Fig. R7d).

Reviewer comments:

5. *The authors might want to refer to these closely related works: Nature Communications 10, 580 (2019); Phys. Rev. Lett. 123, 054301 (2019); Phys. Rev. Lett. 118, 040401 (2017).*

Response:

We thank Reviewer #3 for bringing these nice works to us. The non-Hermitian origin of surface waves presented in these works is a novel type of non-Hermitian bulk-boundary correspondence and is closely related to the topic of our work. **We have cited these works in the revised manuscript as refs. 24, 30 and 31.**

Reviewer comments:

6. *The authors are using analogies between optics and acoustics, and these have been recently (2019 and 2020) explored in depth in the following works: Transverse spin and surface waves in acoustic metamaterials, Phys. Rev. B 99, 020301(R) (2019); Spin and*

Fig. R7. Effective dimerization for topological and trivial lattices. **a**, Unit cell of the non-Hermitian lattice in topological phase, which can be effectively mapped to a lattice with coupling dimerization shown in **b**. **c**, Unit cell of the non-Hermitian lattice in trivial phase, which can be effectively mapped to a lattice with coupling dimerization shown in **d**.

orbital angular momenta of acoustic beams, Phys. Rev. B 99, 174310 (2019); Acoustic versus electromagnetic field theory: scalar, vector, spinor representations and the emergence of acoustic spin, New Journal of Physics 22, 053050 (2020); Edge modes in two-dimensional electromagnetic slab waveguides: Analogs of acoustic plasmons, Phys. Rev. B 102, 045129 (2020). I am not requesting all of these to be cited, but it seems that the authors better to be aware of them, because what they are doing is importing ideas from optics and CM into acoustics, and this is what these recent works also do.

Response:

We thank Reviewer #3 for mentioning these related works. We also feel we should point out to readers that many phenomena can find analogies between these two fields, despite the fact that acoustic wave is longitudinal. We have added one sentence on the right column of page 4 in the main text as "Moreover, recent studies have shown that many phenomena are commonly found in both acoustics and photonics.". Also, related works are included as

refs. 59-62.

Reviewers' Comments:

Reviewer #1:

Remarks to the Author:

The authors have revised the manuscript as required. I am happy to recommend the revised manuscript for publication in Nature Communications.

Reviewer #3:

Remarks to the Author:

The three referees provided a very large number of comments and suggestions, to improve this work. The authors did an excellent job providing very persuasive and good replies to all the issues raised. Excellent job.

This revised manuscript is considerably improved, over the previous version, which was still excellent.

Thus, in my opinion, this revised version is ready for publication in Nature Comm.

Response Letter to Reviewers

We are grateful for the encouraging comments on this manuscript (NCOMMS-20-26559A) from both the reviewers. In the text below the comments from each reviewer are quoted in *blue italics*, and followed by our response.

Reviewer #1

Reviewer comments:

The authors have revised the manuscript as required. I am happy to recommend the revised manuscript for publication in Nature Communications.

Response:

We thank Reviewer #1 for the recommendation.

Reviewer #3

Reviewer comments:

The three referees provided a very large number of comments and suggestions, to improve this work. The authors did an excellent job providing very persuasive and good replies to all the issues raised. Excellent job.

This revised manuscript is considerably improved, over the previous version, which was still excellent.

Thus, in my opinion, this revised version is ready for publication in Nature Comm.

Response:

We thank Reviewer #3 for the recommendation.